# Among Other Tissues, Short-Term Garlic Oral Treatment Incrementally Improves Indicants of Only Pancreatic Islets of Langerhans Histology and Insulin mRNA Transcription and Synthesis in Diabetic Rats

**DOI:** 10.3390/biology13050355

**Published:** 2024-05-18

**Authors:** Amani M. Al-Adsani, Khaled K. Al-Qattan

**Affiliations:** Department of Biological Science, Faculty of Science, Kuwait University, P.O. Box 5969, Safat, Kuwait City 13060, Kuwait; amani.aladsani@ku.edu.kw

**Keywords:** garlic, diabetes, pancreas, Langerhans, insulin, mRNA

## Abstract

**Simple Summary:**

Type 1 diabetes mellitus is a medical condition that is characterized by a decrease in insulin level and an increase in glucose concentration in the body. Diabetes is detrimental to the general health condition, and many people are becoming affected worldwide. In recent years, a considerable drive has begun to use natural products as management and treatment substances for numerous medical conditions, particularly for diabetes mellitus. It was shown that garlic, a natural product, can increase insulin levels in the body; however, the exact source and genetic mechanism by which this effect transpires is not yet clear. The current study investigated the effect of garlic on the pancreas, bile duct, and liver. In response to a short 8-week treatment period, it was observed that garlic was capable of revitalizing insulin synthesis and secretion only in the pancreas and solely by re-activating and neogenesis of cells restricted to the islets of Langerhans. Short-term treatment with garlic did not have a positive effect on the bile duct and liver insulinogenesis. Nonetheless, indications are that with a longer treatment, this effect might also come to pass. It is believed that garlic might be of considerable importance in the future for diabetics as an accessible and low-cost remedy in the management and treatment of diabetes mellitus type 1.

**Abstract:**

Background: The source, mRNA transcription, and synthesis of insulin in the pancreas, in addition to the bile duct and liver, in streptozotocin (STZ)-induced diabetic rats (DR) in response to garlic oral treatment are not yet clear. Objective: This study investigated the accumulative effects of continued garlic oral treatment on changes in the pancreas, bile duct, and liver with regards to: 1—Insulin mRNA transcription, synthesis, and concentration in relation to changes in serum insulin (SI); 2—Insulinogenic cells insulin intensity and distribution, proliferation, and morphology. Method: Fasting blood glucose (FBG) and insulin concentration in serum and pancreas (PI) and sources and mRNA transcription in the pancreas, bile duct, and liver in normal rats given normal saline (NR-NS) and DR given either NS (DR-NS) or garlic extract (DR-GE) before and after 1, 4, and 8 weeks of oral treatment were examined. Results: Compared to NR-NS, DR-NS showed a significant increase in FBG and reductions in SI and PI and deterioration in islets histology, associated pancreatic insulin numerical intensities, and mRNA transcription. However, compared to DR-NS, the targeted biochemical, histological, and genetic variables of DR-GE were significantly and incrementally improved as garlic treatment continued. Insulin or its indicators were not detected either in the bile duct or the liver in DR-GE. Conclusions: 8 weeks of garlic oral treatment is enough to incrementally restore only pancreatic islets of Langerhans insulin intensity and insulinogenic cells proliferation, morphology, and distribution. These indices were associated with enhanced pancreatic insulin mRNA transcription and synthesis. Eight weeks of garlic treatment were not enough to stimulate insulinogenesis in either the bile duct or the liver.

## 1. Introduction

Insulin, a dipeptide hormone, is a potent stimulator of cellular uptake of glucose and anabolism of carbohydrates, lipids, and proteins. Accordingly, insulin profoundly contributes to maintaining the homeostatic level of circulating glucose within 4–6 mM in the body [1]. Insulin is produced by the β-cells of the islets of Langerhans in the pancreas and is mainly released in response to hyperglycemia [2]. Under normal conditions, insulin is released every 3–6 min, resulting in a circulating concentration ranging between 57 and 79 pmol/L [3]. A chronic decline in the concentration or absence of circulating insulin due to causes including autoimmune-induced β-cell hypoplasia leads to sustained hyperglycemia and a subsequent array of linked biochemical and physiological abnormalities that are collectively designated as diabetes mellitus (DM) type 1 [2]. Masses of people worldwide are proclaimed to have DM, and the majority live in low-income countries. Moreover, in these regions, the number of individuals with DM is increasing drastically every decade, putting a huge burden on the health care services and genuine progress indicators. The end stage of DM, in its different forms, eventually leads to the mortality of a substantial number of diabetics, thereby inflicting a substantial impact on humanity [3].

For investigative studies, type 1 DM is induced in rats by intraperitoneally injecting streptozotocin (STZ), a glucosamine-nitrosourea substance. STZ has a potent antineoplastic activity and is selectively toxic to mammalian pancreatic β-cells. Owing to its glucose-like molecular structure and lipophobic properties, STZ is specifically taken up and accumulated in active β-cells expressing active glucose transporter 2 (GLUT2). In these functional β-cells, STZ causes severe DNA alkylation followed by fragmentation and cellular necrosis [4,5]. Moreover, STZ has a half-life of 15–40 min; thus, it is quickly and almost completely cleared from the body within a short time following administration [6]. Under normal conditions or even in experimental setups, pancreatic regeneration after injury has been reported [7]. Part of this pancreatic healing and, therefore, insulinogenic reclaim occurs in response to treatment with herbal and plant extracts [8,9].

Garlic (*Allium sativum* L.) is a natural herbal product, and its different extract preparations contain a myriad of constituents, many of which are sulfur-containing compounds [10,11,12]. These constituents of garlic, individually or synergistically—as part of an extract—were reported to have a multitude of bioactive actions [12,13,14,15,16]. Specifically, garlic ingredients such as allyl propyl disulfide, allicin, cysteine sulfoxide, and S-allyl cysteine sulfoxide were reported to decrease blood glucose levels by enhancing insulin secretion from pancreatic beta cells, isolate insulin from its bonded forms, and/or increase respective cells sensitivities to insulin [15,17].

Additionally, and along the same line of observations, it was reported that insulin-producing cells were induced by the application of different chemical stimulants to the pancreas, in addition to various other organs, including the bile duct and liver [18,19]. This partial functional transdifferentiation of non-insulinogenic entities into insulin-producing cells is possible since the bile duct and liver, as is the pancreas, originate from the anterior endoderm during the early stages of embryonic development. Thus, the bile duct and liver can harbor cells that can be induced to form insulin-producing cells, as demonstrated in various models by non-natural stimulants [20,21,22]. Accordingly, garlic’s insulinogenic action could also be the result of β-like cell neogenesis not only in the pancreas but also in the bile duct and liver, as suggested by Milner [23]. In the pancreas, both endocrine and exocrine cells (acinar or ductal cells) were suggested to have the potential to give rise to insulin-producing cells [7,24]. This functional transdifferentiation could only happen as a result of molecular alternations at the genetic level. In support, exogenous bioactive compounds from a variety of herbs and plants, which are prevalent in many foods, were reported to compel molecular modulations, including the activation of transcription factors involved in β-cell development [9,23,25,26,27,28,29]. Irrespectively, and as recently reported in a review by Sanie-Jahromi et al. [16], the insulinogenic–genetic mechanism of garlic in the pancreas is still not clear.

To better understand this insulinogenic function of garlic, in addition to the alternative organs and cells that might contribute to the rise in SI level in type 1 DM rats, this study investigated the accumulative effect of garlic oral treatment on changes in the pancreas, bile duct, and liver in relation to: 1—Insulin mRNA transcription, synthesis, and concentration in relation to changes to SI level; 2—Insulinogenic cells insulin intensity and their distribution, proliferation, and morphology.

## 2. Materials and Methods

### 2.1. Experimental Animals

The animals used in this study were 124 healthy male *Sprague–Dawley* rats (HR, body weight: 160–180 g, 14–16 weeks old). The original stock of rats was procured from Harlan Laboratories (Indianapolis, IN, USA). These animals were housed in the Animal Care and Breeding Unit under regular ambient conditions (temperature 22–24 °C; humidity 30–35%; natural light/dark cycle) and provided with a standard rodent diet (Special Diet Services, Chelmsford, Essex, UK) and filtered tap water ad libitum. The rats were handled, maintained, and experimented on according to the international guidelines in the Guide for the Care and Use of Laboratory Animals [30]. The experimental work of this study was also approved by The Department of Biological Sciences ‘Ethical Committee for the Use of Laboratory Animals’—Approval Code: DBS/IRB(ECULA)19-003.

### 2.2. Induction of Diabetes

The weight and age of rats used in this study were found to be the most appropriate for the induction of type 1 DM with a negligible rate of mortality. This type of DM was chemically induced in the required number of HRs by intraperitoneally injecting each rat with a single dose of STZ (Merck, Kenilworth, NJ, USA) solution (60 mg/kg in 0.5 mL 0.01 M sodium citrate buffer, pH 4.5) after 2 h of fasting. Additionally, a control group of normal rats was prepared by intraperitoneally injecting the required number of HRs with a single dose of 0.5 mL of citrate buffer, as previously described by our group [31].

The physical, physiological, and biochemical characteristics of the NR (injected with citrate buffer) and DR (injected with STZ solution) were typical of those of HR and DR, respectively. The physical and biochemical characteristics of these animals are delineated further in the Section 3.

### 2.3. Quantification of Fasting Blood Glucose (FBG)

FBG of all rats was measured using a drop of tail blood and a portable glucometer (OneTouch UltraEasy; LifeScan, Milpitas, CA, USA) after overnight fasting on day 6 post-injection (basal level = BL) of either STZ or citrate buffer. FBG was also measured at 1, 4, and 8 weeks after oral treatment. At BL, the STZ-injected rats with FBG ≥ 16 mmol/L were considered type 1 diabetic rats [DR, n = 80], whereas citrate buffer-injected rats with FBG ≤ 7 mmol/L were considered normal rats (NR, n = 44). The measurements of FBG indicated that the STZ-injected rats were diabetic 6 days post-injection. This STZ diabetes induction efficacy is as previously reported by our group [32]. Furthermore, in a review by Deeds et al. [33], it was stated that diabetes ensued in rats injected with STZ as soon as two days post-injection.

### 2.4. Preparation of Garlic Aqueous Extract

Garlic (*Allium sativum* L., purchased from the local market) aqueous extract (GE) was prepared, stored, and used as previously described by our group [34]. Briefly, GE was prepared as follows: Garlic bulbs were peeled on crushed ice. Then, 50 g of the peeled garlic was cut into small pieces and homogenized in 70 mL of cold, sterile 0.9% NaCl in the presence of some crushed ice. The homogenization was carried out in a blender at high speed using 30 s bursts for a total of 10 min. The homogenized mixture was filtered 3 times through a cheesecloth, the filtrate was centrifuged at 2000 RCF for 10 min, and the clear supernatant was diluted to 100 mL with normal saline. The concentration of this garlic extract (GE) preparation was considered to be 500 mg/mL on the basis of the weight of the starting material (50 g/100 mL). GE preparation was divided into suitable aliquots for one-time, daily use and stored at −20 °C.

Gas chromatography and mass spectrometry analysis showed that the GE preparation was stable after different durations of storage at −20 °C, as reported previously by our group [35]. The used garlic specimen is kept at Kuwait University Herbarium (KUH) under specimen number: ALNAQEEB MA/001/1998 KTUH.

### 2.5. Oral Treatment with Normal Saline or GE

The designated NR (n = 36) were treated with normal saline (NS; NR-NS, 0.5 mL/kg). Alternatively, the designated DR (n = 72) were divided into two subgroups: one group (n = 36) was treated with NS (DR-NS, 0.5 mL/kg), and another (n = 36) was treated with GE [DR-GE, 500 mg/(mL·kg)]. Treatment with either NS or GE was performed at mid-morning as a single daily oral dose and continued for each subgroup of rats, as shown in Figure 1. One set of NR (n = 8) and one set of DR (n = 8) did not receive any treatment, and their data were used to determine the BL of the targeted biophysical, physiological, and biochemical parameters (Figure 1).

### 2.6. Determination of Body Weight, Food and Water Intake, and Urine Output

The body weight (B.Wt.), food (F.I.) and water (W.I.) intake, and urine output (U.O.) of each rat were estimated gravimetrically and volumetrically using metabolic cages at day 7 post-injection (BL—no treatment administration, NR [n = 8] + DR [n = 8]) and at the end of week 1 (W1, NR-NS [n = 12] + DR-NS [n = 12] + DR-GE [n = 12]); week 4 (W4, NR-NS [n = 12] + DR-NS [n = 12] + DR-GE [n = 12]); and week 8 (W8, NR-NS [n = 12] + DR-NS [n = 12] + DR-GE [n = 12]) (Figure 1).

### 2.7. Sacrifice of Rats and Collection of Samples

Following anesthesia (0.2 mL/100 g body weight) with a mixture of ketamine (9 mL, 10%; Dutch Farm International, Nedarhorst den Berg, The Netherlands) and xylazine (1 mL, 10%; Interchemie, Venray, The Netherlands) and under aseptic conditions, each rat was secured on its dorsal side, and the thoraco-abdominal cavities were exposed using heavy-blade scissors. The targeted internal organs were carefully manipulated; thereafter, the required samples were collected. This procedure was carried out after each animal’s full sedation and following the completion of its designated treatment period [BL, W1, W4, and W8 (Figure 1)].

### 2.8. Determination of Serum Insulin (SI) Concentrations

Blood samples were collected via cardiac puncture from each rat at BL and at the end of the designated treatment periods (Figure 1). The samples were allowed to stand at room temperature (22–24 °C) for 30 min. Subsequently, serum was collected and stored at −20 °C and later analyzed for SI concentrations using ELISA kits and following the manufacturer’s instructions (SPI bio/BioAlert Solutions, Sherbrooke, QC, Canada).

### 2.9. Immunohistochemical Localization of Insulin in the Pancreas, Bile Duct, and Liver; Pro-Insulin and C-Peptide Determination in the Bile Duct and Liver; and Insulin Intensity Estimation in the Pancreas

Pancreatic, bile duct, and liver tissue samples were collected at BL and the end of each designated treatment period (Figure 1) and later analyzed for insulin immunohistochemical localization and intensity. Each sample was washed in PBS (pH 7.4; Sigma-Aldrich, St. Louis, MO, USA) and fixed overnight in 4% paraformaldehyde (Merck, Rahway, NJ, USA). The samples then underwent histological preparation and embedding into paraffin wax (Histo-embedder; Leica, Wetzlar, Germany); 4-μm sections were cut (5 μm sections were used for Z-stack images), dewaxed, rehydrated, and permeabilized with 1% Triton X-100 (Sigma-Aldrich). Antigen retrieval was performed using Tris-EDTA (Sigma-Aldrich) for 60 min. To prevent endogenous peroxidase activity, tissue sections were blocked using a peroxidase-blocking solution (Envision peroxidase system; Dako, Glostrup, Denmark). Additional blocking with 2% blocking buffer (Roche, Basel, Switzerland) in PBS was carried out to inhibit non-specific binding. Different tissue sections were incubated overnight with rabbit anti-insulin-1 antibody (dilution ratio: 1:400—Abcam, Cambridge, UK), rabbit anti-C-peptide antibody (dilution ratio: 1:100—Cell Signaling Technology, Danvers, MA, USA), or mouse anti-insulin/pro-insulin antibody (dilution ratio: 1:400—Invitrogen, Carlsbad, CA, USA) diluted in blocking buffer at 4 °C. After several washes with PBS, tissue sections were incubated with species-specific peroxidase-labeled secondary antibody. Tissue signals indicating insulin presence were detected using the DAB substrate-chromogen kit specific for mice or rabbits (Dakocytomation Envision System/HRP, Dako), followed by counterstaining with hematoxylin and mounting. Tissue sections were examined using a light microscope equipped with a camera (ECLIPSE Ti2-E; Nikon, Tokyo, Japan), and representative images were captured. Only pancreatic tissue showed signs of insulin presence; thus, the intensity of insulin immunolocalization was estimated by analyzing three images from three slides for three rats at each designated time of sacrifice for each group using NIS-Elements BR4.6 (Nikon) following Nikon image analysis procedures. For confirmation, the bile duct and liver tissues were further tested for pro-insulin and C-peptide (dilution ratio: 1:100) to detect any indicators of insulin synthesis or presence.

### 2.10. Determination of Pancreatic Insulin (PI) Concentration

Pancreatic tissue samples were collected at BL and at the end of each designated treatment period (Figure 1). They were stored at −20 °C and later analyzed for insulin concentrations using ELISA kits and following the manufacturer’s instructions (dilution ratio: 1:400—SPI bio/BioAlert Solutions).

### 2.11. Determination of PI mRNA Transcription

Pancreatic tissue samples were collected at BL and at the end of each designated treatment period (Figure 1). Total RNA was extracted using the RNeasy Plus Universal Mini Kit (Qiagen, Hilden, Germany) and following the procedure modified by Aladsani et al. [36] and stored at −80 °C. RNA quality (integrity) was measured with Agilent RNA 6000 Pico-Kit (Chips, Santa Clara, CA, USA) using Agilent 2100 Bioanalyzer (Life Technologies, Carlsbad, CA, USA). Reverse transcription of mRNA to cDNA was performed using the SuperScript IV First-Strand Synthesis System (Life Technologies). The housekeeping gene *β-actin* was used as an internal control, and insulin mRNA transcription was determined through qRT-PCR using a TaqMan gene expression assay with the labeled probes, *β-actin* [code no. Rn00667869 (VIC)] and *INS2* (code no. RN01774648 g1; FAM), following the manufacturer’s instructions (Applied Biosystems, Waltham, MA, USA).

### 2.12. Pancreatic Amylase and Insulin Co-Localization in the Pancreas

Since only the pancreas revealed signs of insulin presence; thus, the co-localization of amylase (indicative of exocrine entities) and insulin (indicative of endocrine entities) was investigated by co-labeling the pancreatic tissue with rabbit anti-insulin/pro-insulin antibody (dilution ratio: 1:400—Invitrogen) and mouse anti-amylase (dilution ratio: 1:100—Santa Cruz Biotechnology, Dallas, TX, USA). The following day, after several washes with PBS, species-specific secondary antibodies were added. Texas Red goat anti-rabbit (dilution ratio: 1:100—Sigma-Aldrich, Gillingham, UK) and FITC goat anti-mouse (dilution ratio: 1:100—Sigma-Aldrich, Gillingham, UK) were both prepared. The slides were left in the dark for 1 h before washing three times with PBS and mounting with DAPI (Abcam). Sections were analyzed using an ECLIPSE Ti2-U inverted microscope (Nikon, Tokyo, Japan) and NIS-Elements BR4.6 software (Nikon, Tokyo, Japan) for screening, and images were collected on an A1 confocal microscope (Nikon) and NIS-Elements version 4.5 (Nikon).

### 2.13. Data Presentation and Statistical Analyses

B.Wt., F.I., W.I., U.O., FBG, SI and PI concentrations, and insulin mRNA and immunohistochemistry (IHC) numerical intensities are presented in bar graphs as mean ± standard error of the mean. Images of the PI-IHC are collated into three sets of images as ×4 illustrations. Images of amylase and insulin co-localization are presented. The readings were compared using two-way analysis of variance with a least-significant-differences post hoc test (SPSS v.22; IBM, Armonk, NY, USA). Statistical significance was established at *p* < 0.05.

## 3. Results

### 3.1. B.Wt., F.I., W.I., and U.O

The BL of B.Wt., F.I., W.I., and U.O of NR-NS were normal and comparable to those observed in healthy rats. Alternatively, at BL and before recommencing any treatment, the two diabetic groups compared to the NR-NS group showed similar B.Wt. but a significantly elevated F.I., W.I., and U.O. (Figure 2A–D). During the 8 weeks of the study, the NR-NS group demonstrated significant continuous increases in B.Wt., reaching a maximum gain of 130% (Figure 2A) and a constant F.I. (Figure 2B), W.I. (Figure 2C), U.O. (Figure 2D) compared with their BL. Conversely, the DR-NS group showed a continuous decrease in B.Wt. culminating in a significant maximal loss of 44% compared with their BL and 360% less than the B.Wt. of the NR-NS group at W8 (Figure 2A). Considering the values at BL and those of the NR-NS group, DR-NS consumed 60% more food (Figure 2B), drank 450% more water (Figure 2C), and excreted 718% more urine (Figure 2D). These elevated levels of intake and output remained steady during the subsequent three stages of treatment at W1, W4, and W8. Conversely, the DR-GE group showed a nearly typical healthy behavior and a trend of gradual increase in B.Wt., which became significantly higher, by 43%, than their BL weight and 136% higher than the B.Wt. of the DR-NS group at W8 (Figure 2A). Furthermore, the DR-GE group showed an initial insignificant trend of a gradual decrease in F.I. (Figure 2B) and W.I. (Figure 2C); however, both parameters became significantly lower, by 29 and 32%, than their own BLs and those of the DR-NS group at W8, respectively (Figure 2B,C). The daily U.O. of the DR-GE group (Figure 2D) was significantly lower, by 15 and 19%, at W4 and W8, respectively, than the DR-NS values.

### 3.2. FBG and SI Concentrations

The BL of FBG and SI of NR-NS were normal and similar to those of healthy rats. Alternatively, at BL and before receiving any treatment, the two diabetic rat groups, compared to the NR-NS group, showed a significantly higher FBF and lower SI (Figure 3A,B). The mean FBG and SI concentrations of the NR-NS group during the successive stages of treatment were steady at 7.5 mmol/L and 1.02 pg/mL, respectively (Figure 3A,B). The DR-NS group showed significantly higher FBG levels, by 238%, than those of the NR-NS group and their BLs. Hyperglycemia in the DR-NS group increased by an additional 38% at W8 compared with BLs (Figure 3A). Conversely, the SI concentration of the DR-NS group was significantly lower, by 468%, at BL and than that of the NR-NS group. The SI concentration of the DR-NS group decreased by an additional 34% at W8 compared with the BL (Figure 3B). The FBG of the DR-GE group decreased significantly and progressively as the treatment with GE continued compared with the values of the DR-NS group and were similar to those of the NR-NS group at W8 (Figure 3A). The continuous hypoglycemic action of GE in the DR-GE group was paralleled by continuous elevations in SI concentration (reaching as high as 2352% increase) compared with the values of the DR-NS group at W8 and as high as 409% compared with the BL in the DR-GE group (Figure 3B).

### 3.3. Pancreas, Bile Duct, and Liver Insulin; Immunohistochemical Localization and mRNA Transcription; and Pro-Insulin and C-Peptide Indicators

The BL of PI concentration and mRNA transcription of the NR-NS group were normal and as can be detected in healthy rats. Alternatively, at BL and before administering any treatment, the two diabetic rat groups, compared to the NR-NS group, showed a significantly lesser PI concentration and mRNA transcription (Figure 3A,B). The expression of PI mRNA in the NR-NS was consistent during the four stages of the experiment, with an average value of 1.0 ∆∆Ct. In the DR-NS group, the expression of PI mRNA at the BL was significantly lower, by 99%, than that in the NR-NS group. Furthermore, the expression of PI mRNA in the DR-NS group decreased by an additional 70% at W8. The expression of PI mRNA in the DR-GE group showed an initial decline, which subsequently increased by 50% at W4 and by a further 67% at W8 compared with its BL (Figure 3C).

In the NR-NS group, the measured PI concentrations were steady during the four stages of the experiment at an average value of 772 pg/g tissue. Alternatively, in the DR-NS group, the BL value of the PI was significantly lower, by 86%, and decreased even further by 55% at W8. Oppositely, in the DR-GE group, there was a continuous accumulation in the PI concentration (reaching as high as 243%) and PI-IHC. This increase in PI indicators generally followed the same trend of progressive and significant increase in pancreatic mRNA, especially at W8 in this group of treated rats (Figure 3D).

Immunohistological analysis of the bile duct and liver for markers of insulin, pro-insulin, and C-peptide did not reveal any evidence even after 8 weeks of GE treatment (Figure 4). Alternatively, treating DR with GE led to significant and progressive increases in the targeted attributes of insulin in the pancreas, as detailed hereafter.

Considering the IHC localization of PI, the images of the NR-NS group showed typical distribution (Figure 5A) and numerical intensities (Figure 5B) of insulin during the four stages of monitoring (BL, W1, W4, and W8). The BL of the NR-NS group shows normal IHC localization of insulin restricted to the islets of Langerhans. Alternatively, the image in the DR-NS group showed considerably decreased IHC localization (Figure 5A) and numerical intensity (Figure 5B) of insulin, which declined even further toward the end of the study. Furthermore, in this group, there was a decline in the islets of Langerhans population, which was associated with a considerable drop in the count of endocrinal cells with a loss of typical morphology. Alternatively, in the DR-GE group, although the image of BL IHC showed minimal evidence of insulin localization and distribution as that observed at the same stage in the DR-NS group, as treatment with GE commenced and continued, the DR-GE group showed progressively increased islets of Langerhans typical distribution and insulin-tagged cells (Figure 5A) with elevated numerical intensities (Figure 5B). In the DR-GE group, the distribution and prominence of the islets of Langerhans were almost similar to those observed in the NR-NS group.

Testing for pancreatic amylase and insulin localization in pancreatic samples of DR-GE collected at W8 showed that insulin expression (tagged red) was limited to the (endocrinal portion) islet of Langerhans. Alternatively, the acinar cells (exocrine portion) only revealed the presence of amylase (tagged green) without any co-localization with insulin (Figure 5C). Furthermore, a confocal Z-stack image of a pancreas from the DR-GE group (Figure 5D) showed a typical and normal proliferation of endocrinal cells with normal morphology within the islets of Langerhans compared with that observed in the pancreas of the NR-NS group.

## 4. Discussion

Type 1 DM is a debilitating metabolic and physiologic abnormality. It results from a reduction in the concentration or absence of circulating insulin due to a decline in the mass and function of pancreatic β-cells. Managing and treating type 1 DM involves following special dietary protocols and lifestyle along with the administration of antidiabetic medication, which in advanced cases includes injections of exogenous insulin. Cell replacement therapy, such as islet of Langerhans transplantation, is another management approach; however, it is limited by donor availability and histocompatibility [37]. Owing to the inconveniences and limitations of the currently adopted pharmaceutical therapies, many studies have investigated alternative management protocols for DM, such as β-cell neogenesis [38], which is induced by certain synthetic or natural chemical stimulations. Considering our findings and those of previous reports, oral administration of GE for 8 weeks attenuated the effects of DM on FBG, B.Wt., F.I., W.I., and U.O. These ameliorative effects of garlic were suggested to be mediated via elevating insulin circulating concentration, making this hormone more readily available to its target cells [39]. In this study, it was observed that the progressive amelioration of some of the biochemical and physiological symptoms of DM was positively and incrementally correlated with the increase in insulin circulating concentration in the DR-GE during the three successive stages of oral intake of garlic. To investigate the source(s) of insulin causing the increase in SI concentration in the DR-GE group, we targeted the pancreas, bile duct, and liver in these rats. Moreover, our study employed a time-related investigation of insulin mRNA transcription and producing cell location. We also explored the presence of pro-insulin and C-peptide in the bile duct and liver to explore the possibility that these two organs might also be involved in and contribute to insulin production in STZ-induced type 1 DM rats as a result of GE treatment.

Although the single dose of STZ administered should have destroyed most of the β-cells within the pancreas of the DR, as can be deduced from the IHC data of the DR-NS group, this organ still represented a strong potential target for investigating the insulinogenic action of GE. The pancreas possesses endocrine self-renewal and expansion capabilities during “normal” physiological conditions, including pregnancy and obesity [40,41]. Additionally, the plasticity of the pancreatic cells exists through their ability to regenerate following injury or gene induction [42]. In this study, the accumulative increments in the SI concentration in response to GE treatment positively correlated with corresponding gene activation, which most likely can be attributed to increased tissue mRNA transcription and insulin localization, intensity, and distribution in the pancreas. Furthermore, this pancreatic reclaim of its insulin synthesis capability and secretagogue might be a result of the presence of immature β-cells that evaded the destructive actions of the single dose of STZ administered to induce diabetes and later became mature or activated by GE treatment. In support, Van der Meulen et al. [43] showed that the periphery of islets of Langerhans has a neogenic niche for transcriptionally immature β-cells that lack GLUT-2; therefore, they cannot sense glucose, or STZ for that matter. These so-called “virgin” β-cells may become functionally mature in response to the right stimulus, such as garlic, as was observed when comparing the IHC data of the DR-GE and DR-NS in this study. Pancreatic progenitor cells could also be insulin-producing cells. Putative progenitor cells have been shown to exist in various areas of the pancreas, including the intra-islets region, which reportedly contribute to β-cell regeneration [44]. However, our confocal microscopy data support the possibility that GE stimulated only the activation and/or differentiation of most likely the “virgin” β-cells because insulin tagging was observed only in the islets of Langerhans. This is supported by the increased numerical and IHC intensity of insulin in the DR-GE group. Our current findings do not support the idea that other pancreatic exocrine cell types (such as acinar or centroacinar cells) could have transdifferentiated into insulin-producing β-like cells. In a previous study, exocrine transdifferentiation was induced by modulations of ambient temperature [45]. In this study, the regeneration of pancreatic β-like cells in the DR-GE group without induction by garlic is rather unlikely as the pancreases of the DR-NS group showed continuous deterioration of insulin indicators and cell size and morphology within the islets of Langerhans.

Insulin-producing and -releasing β-like cells have been found in other tissues and organs [46]. These other tissues and organs include the biliary system and liver, which are derived from the same embryonic layer as the pancreas. The liver was reported to have the potential to regenerate after injury from progenitor cell population or oval cells [47]. Liver cells also possess GLUT2 transporters, which are characteristic of β-cells. Ferber et al. [20] demonstrated the possibility of converting mouse liver cells into insulin-producing cells through genetic manipulation. Since then, other researchers have also managed to produce β-like cells from the liver [22,48]. Additionally, the common bile duct of the liver may naturally harbor β-cells or cells that can be induced to become insulin-producing cells [49]. Accordingly, we extracted and investigated both the bile duct and liver of DR-GE rats for actual indicators of insulin, pro-insulin, and C-peptide. None of these markers were detected in these two tissues. Our current observations regarding the bile duct and liver are inconsistent with those of Kim et al. [50], Vorobeychik et al. [51], and Preisegger et al. [52], who, notably, used stimulants other than garlic. We strongly believe that a longer treatment period with garlic than that used in this study might be needed for non-insulin-producing bile duct and liver cells to differentiate into insulin-producing β-like cells.

A garlic-stimulated pancreatic insulin secretagogue pathway is not clear, as stated in a very recent and comprehensive review by Sanie-Jahromi et al. [16]. Previous studies have gone as far as reporting that garlic augments circulatory insulin concentration [13,53], elevates pancreatic insulin histochemical detection [10,54], and increases non-insulin-related gene expressions [16]. In the current study, GE was found to increase the transcription of insulin mRNA only in the pancreas but neither the bile duct nor the liver of DR-GE. This upregulation in insulin mRNA could have been a consequence of an increase in the expression of three key transcription factors involved in β-cell differentiation, which are *Ngn3*, *Pdx1*, and *MafA*, as we reported recently [29]. Furthermore, this action of GE on insulin mRNA transcription was found to be incremental as treatment continued. Based on observations from this and earlier studies, it can be postulated that GE could stimulate pancreatic cells as a result of sequential increases in: 1—transcription factors, such as *Ngn3*, *Pdx1*, and insulinogenesis *MafA*, which are involved in β-cell differentiation; 2—number and size of insulin-producing cells only in the islets of Langerhans; 3—insulin mRNA transcription and translation; 4—pancreatic insulin synthesis; 5—serum insulin concentration (Figure 6).

Being a detrimental medical condition that in severe cases leads to fatality, it is imperative to manage and control type 1 DM by implementing any curative procedure that is possible, including the use of natural substances. The observation that natural products, such as garlic, have ameliorative actions on many symptoms of type 1 DM; it is suggested and proclaimed that garlic might be an essential supplementary medication [16]. In a recent review, Yedjou et.al. [55] have once again supported and asserted the use of garlic, among other natural herbs and vitamins, as chemo-therapeutic/preventive agents for the management of DM. This potential of garlic is based on the observations and findings of many studies, including this one, that garlic corrects abnormalities in the physiological mechanisms and biochemical pathways of type 1 DM.

## 5. Conclusions

The findings of this study suggest that: (1) The insulinogenic properties of garlic in the pancreas are accumulative, which is a remedial characteristic similar to that of synthetic pharmaceutical agents; (2) The insulinogenic action of garlic is partially a consequence of augmented insulin mRNA transcription and synthesis; (3) Stimulated PI-producing cell regeneration is restricted to the islets of Langerhans; and (4) an 8-week treatment with garlic might not be sufficient to stimulate either the bile duct or the liver to produce insulin. (5) The findings of this study provide further insights into developing natural strategies for managing and treating type 1 DM.

## Figures and Tables

**Figure 1 biology-13-00355-f001:**
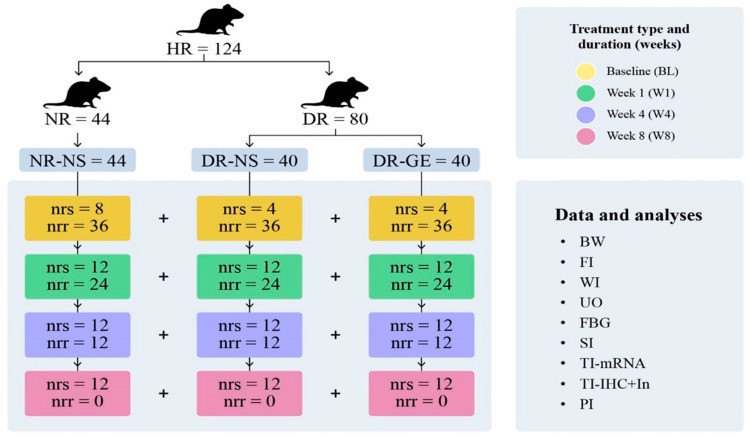
Experimental Protocol. This figure shows the different groups of rats tested in this study: normal rats treated with normal saline (NR-NS), diabetic rats treated with normal saline (DR-NS), and diabetic rats treated with garlic extract (DR-GE). The figure also shows the collected biophysical readings and biochemical, histological, and image analyses carried out at the end of each period of treatment (BL, W1, W4, and W8). These readings include body weight (BW), food intake (FI), water intake (WI), urine output (UO), fasting blood glucose (FBG), serum insulin (SI), tissue insulin mRNA transcription (TI-mRNA), tissue insulin immunohistochemical localization and numerical intensity and confocal microscopy localization (TI-IHC+In), pancreatic insulin (PI), healthy rats (HR), normal rats (NR) injected with citrate buffer, and diabetic rats (DR) injected with streptozotocin (STZ). The number of rats in each group is indicated by the number of rats that were sacrificed (nrs) and number of rats that remained and continued the treatment to the next stage (nrr). Rats used for estimating BL readings did not receive any treatment and were sacrificed six days after injection of either citrate buffer (=4 + 4) or STZ (=8).

**Figure 2 biology-13-00355-f002:**
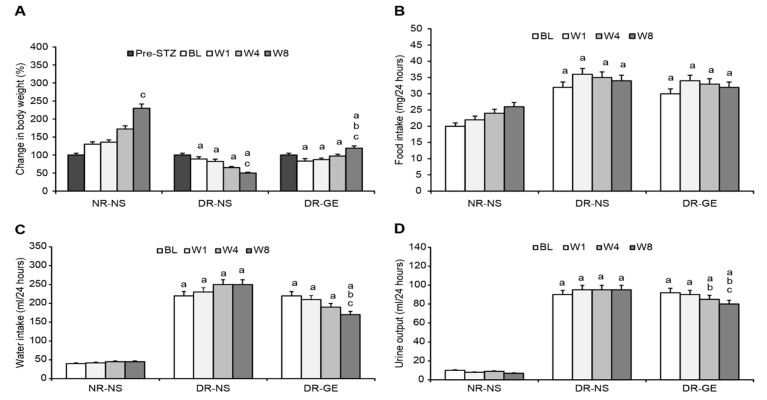
Effect of GE on body weight, food intake, water intake, and urine output. (**A**) The body weight of the DR-GE group increased gradually since the first week and compared to the body weight of the DR-NS group at W8; (**B**) Food intake of the DR-GE group showed an overall insignificant increase; (**C**) Water intake of the DR-GE group showed a significant continuous gradual decline; (**D**) Urine output of the DR-GE group decreased with treatment. Pre-STZ, pre-streptozotocin injection; BL, basal level = day 7 post-STZ injection = pre-treatment; W1, week 1; W4, week 4; W8, week 8; NR-NS, normal rats treated with normal saline; DR-NS, diabetic rats treated with normal saline; DR-GE, diabetic rats treated with garlic extract; GE, garlic extract. a: significantly different compared to NR-NS at the same stage; b: significantly different compared to the DR-NS group at the same stage; c: significantly different compared to BL. *p* ≤ 0.05.

**Figure 3 biology-13-00355-f003:**
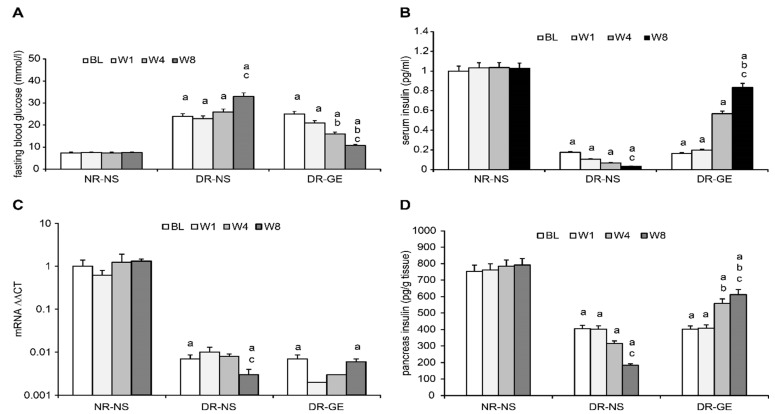
Effect of GE on fasting blood glucose, serum insulin, pancreatic insulin mRNA, and pancreatic insulin concentration. (**A**) FBG of the DR-GE group decreased continuously; (**B**) SI of the DR-GE group showed substantial increases at the last two stages of measurements, although it was only significantly higher compared with its own BL and the value of the DR-NS group at W8 of treatment; (**C**) mRNA values of the DRNS group showed a significant continuous decline. The mRNA values of the DR-GE group showed a certain trend that started with an initial decrease at BL followed by a gradually significant increase. The changes in the values of pancreatic insulin (PI) and mRNA of the DR-GE group were consistent with values of PI, immunohistochemistry (IHC) localization, and protein intensities; (**D**) PI of the DR-GE group showed significant continuous increases compared with the values of the DR-NS group. BL, basal level = Day 7 post-STZ injection = pre-treatment; W1, week 1; W4, week 4; W8, week 8; NR-NS, normal rats treated with normal saline; DR-NS, diabetic rats treated with normal saline; DR-GE, diabetic rats treated with garlic extract; FBG, fasting blood glucose; SI, serum insulin; GE, garlic extract. a: significantly different compared to the NR-NS group at the same stage; b: significantly different compared to the DR-NS group at the same stage; c: significantly different compared to BL. *p* ≤ 0.05.

**Figure 4 biology-13-00355-f004:**
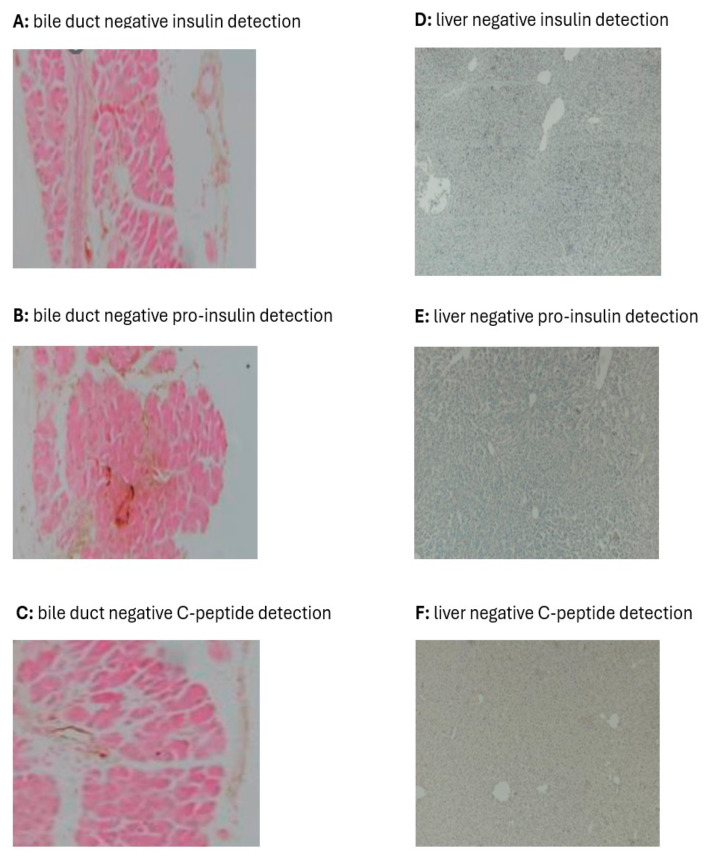
Effect of GE on insulin (**A**,**D**), pro-insulin (**B**,**E**), and C-peptide (**C**,**F**) in the bile duct and liver of diabetic rats. The immunohistochemical images of the bile duct and liver of DR-GE did not show any signs for insulin, pro-insulin, and C-peptide even after 8 weeks of treatment with the GE.

**Figure 5 biology-13-00355-f005:**
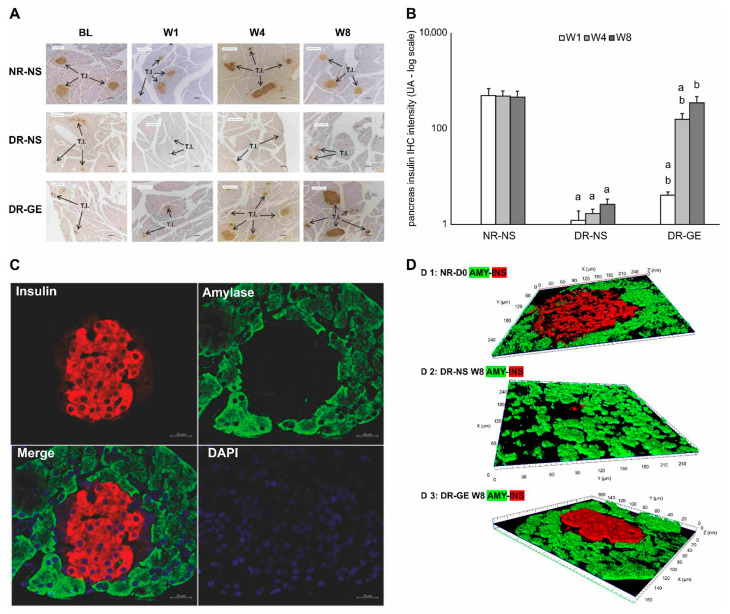
Effect of GE on PI-IHC localization, PI-IHC numerical intensity, and pancreatic co-localization of amylase and insulin. (**A**) Pancreases of the NR-NS group showed an evident and typical distribution of insulin tagging (I.T.). Alternatively, pancreases of the DR-NS group showed a considerably reduced distribution and scarcity in insulin tagging. Compared with the BL of the DR-GE group, pancreatic tissue showed a gradual increase in the distribution and intensity of insulin tagging. (**B**) DR-GE pancreatic IHC insulin numerical intensities showed a significant continuous increase compared with the values of the DR-NS. (**C**) Red color indicates the presence of insulin (**top left**), whereas the green color indicates the presence of amylase (**top right**). When the insulin and amylase images are merged, the resulting image (**lower left**) shows the distinct location of insulin (in the endocrine portion) and amylase (in the exocrine portion) in the pancreas. The lower right image shows pancreas tissue stained blue with DAPI. (**D**) Three-dimensional image of the islets of Langerhans and surrounding tissues; (**D 1**) An image of the pancreas of a representative normal rat at day 0 showing amylase (green) and insulin (red) tagging; (**D 2**) An image of the pancreas of a representative diabetic rat at week 8 treated with normal saline showing amylase and insulin tagging; (**D 3**) An image of the pancreas of a representative diabetic rat at week 8 treated with garlic extract showing amylase and insulin tagging. BL, basal level = day 7 post-streptozotocin = pre-treatment; Different periods of treatment = W1, week 1; W4, week 4; W8, week 8; NR-NS, normal rats treated with normal saline; DR-NS, diabetic rats treated with normal saline, DR-GE, diabetic rats treated with garlic extract; T.I., tagged insulin (magnification = 4×, scale bar = 100 μm); GE, garlic extract; DAPI, 4′,6-diamidino-2-phenylindole; PI-IHC, pancreatic insulin-immunohistochemistry; a: significantly different compared to NR-NS at the same stage; b: significantly different compared to DR-NS at the same stage. *p* ≤ 0.05.

**Figure 6 biology-13-00355-f006:**
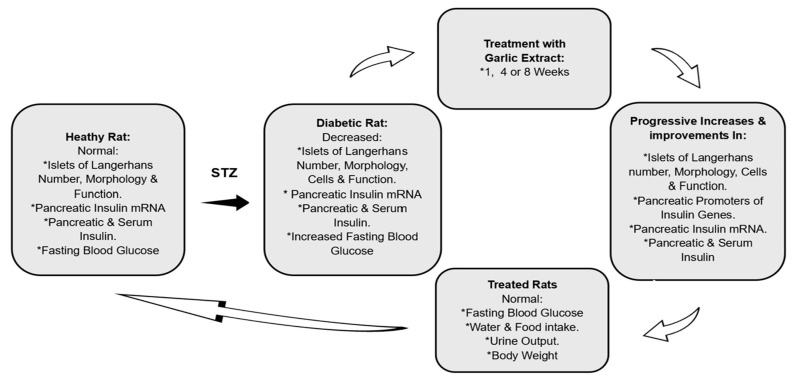
Proposed garlic insulinogenic mechanism in the pancreas and general antidiabetic effects. *: marks different variables.

## Data Availability

All final data acquired are presented in this manuscript. Raw data can be received from the corresponding author upon a written request (khaled.alqattan@ku.edu.kw).

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
