# Peer review of "Among Other Tissues, Short-Term Garlic Oral Treatment Incrementally Improves Indicants of Only Pancreatic Islets of Langerhans Histology and Insulin mRNA Transcription and Synthesis in Diabetic Rats"

_biology, 2024, doi:10.3390/biology13050355_

Round 1

Reviewer 1 Report

Comments and Suggestions for Authors

Overall, it seems like a great job. Your research topic addresses an international issue, and any study on the underlying mechanisms of type 1 diabetes with good methodology will be appreciated. Specifically, there are some comments and observations I would like to express.

An article without an abstract?

Key words?

Line 50. An abrupt change of subject, they could use a connector ("on the other hand", for example).

Review the writing style of the introduction or send the document to a text style editing service. There are small aspects in the writing and errors that make reading difficult (such as in line 79, where a large space was left before the name of an author). Line 87. You can simply call them 'animals and housing conditions'.

Why use rats of such age?

For studying type 1 diabetes, wouldn't it be ideal to use young adult rats?

Providing tap water could affect the quality of your experiments; in a certified vivarium, sterilized water should be available.

In the ethical aspects, you should mention the method of euthanasia at the end of the experiment and the endpoint at which they would be euthanized in case of potential suffering of the animals.

I see that in section 2.7 they included the sacrifice method, heavy blade scissors = guillotine? I ask because you collected blood by cardiac puncture, and if you used a guillotine, a larger volume of blood could have been collected via the neck.

You do not specify why the animals' size is so large, in terms of statistical power and ethics.

The figures appear to be screenshots, they have low resolution and lack sharpness.

At the bottom of Figures, the p-value should be stated.

I consider that the names of the experimental groups should be simpler, in the sense that when analyzing the graphs, it is initially difficult to identify them (for example, the control group could simply be called 'control,' and another group 'Diab+Gar,' etc.)

“Figures speak louder than a thousand words”. I think you should write your results in a much more concise manner, in a more practical sense; in the sense that all the graphs (once one understands the group nomenclature) are clear in trends and differences in variables between groups

The information at the bottom of the figures is very extensive and sometimes repeats what is written in the results section overall. Be more concise

Figure 5, which proposes the mechanisms of action, is at times very optimistic, in the sense that the arrows indicate a path back to 'normality

The discussion seems comprehensive in terms of the physiological mechanisms found in this study, as well as previously. However, I feel that there is a lack of discussion on whether specific compounds present in your extract (such as aloin, etc.) have these effects per se, and suggesting which phytochemicals or group of phytochemicals are specifically responsible for that effect (something mentioned in the introduction but needs further discussion). On the other hand, I did not see any mention of the limitations and prospects of this study. 

Type 1 diabetes is diagnosed at early stages of life, leading to a shorter lifespan even with insulin treatment, diet, etc. In this context, I would like to ask you to mention in the discussion, in what context garlic extract could help patients, as an adjuvant? as complementary medicine, treatment for short or long periods? as a nutraceutical? as a pharmacological combination? Discuss. 

Comments on the Quality of English Language

The English is understandable, but it requires style revision.

Author Response

See attached file: Response to Editorial Office, Reviewers' Comments and Suggestions. 

Reviewer 2 Report

Comments and Suggestions for Authors

The manuscript titled " Among Other Tissues, Short-Term Garlic Oral Treatment Incrementally Improves Indicants of Only Pancreatic Islets of Langerhans Histology, Insulin m-RNA Expression, and Synthesis in Diabetic Rats" demonstrates the beneficial effects of garlic extract on diabetes induced by STZ. The authors observed improvements in physiological parameters in STZ-induced diabetic mice treated with garlic extract, particularly noting an increase in insulin gene expression in pancreatic beta cells. While commendable, the study lacks investigation into the molecular mechanisms underlying these effects.

Further considerations for enhancing the manuscript include:

1.    The authors noted the absence of insulin, pro-insulin, and C-peptide induction in the bile duct and liver but did not provide corresponding data. Including this data in the manuscript would strengthen its credibility and relevance.

2.    Clarification is needed regarding the preparation of experimental models, specifically regarding the injection of STZ or 0.01M sodium citrate buffer into NR mice. It should be clearly stated whether the pre-STZ mice in both the NR-NS and DR-NS groups served as physiologically healthy control mice. Clear delineation in both the figure and the manuscript would improve understanding.

Allover, its a good work.

Author Response

See attached file: Response to Editorial Office and Reviewers Comments and Suggestions
